# Fast Imagic: Solving Overfitting in Text-guided Image Editing via Disentangled UNet with Forgetting Mechanism and Unified Vision-Language Optimization

**Shiwen Zhang**[*]
Bytedance Inc
shiwen.zhang@bytedance.com

## Abstract

Text-guided image editing on real or synthetic images, given only the original image itself and the target text prompt as inputs, is a very general and challenging task. It requires an editing model to estimate by itself which part of the image should be edited, and then perform either rigid or non-rigid editing while preserving the characteristics of original image. Imagic, the previous SOTA solution to text-guided image editing, suffers from slow optimization speed, and is prone to overfitting since there is only one image given. In this paper, we design a novel text-guided image editing method, Fast Imagic. First, we propose a vision-language joint optimization framework for fast aligning text embedding and UNet with the given image, which is capable of understanding and reconstructing the original image in 30 seconds, much faster and much less overfitting than previous SOTA Imagic. Then we propose a novel vector projection mechanism in text embedding space of Diffusion Models, capable of decomposing the identity similarity and editing strength thus controlling them separately. Finally, we discovered a general disentanglement property of UNet in Diffusion Models, i.e., UNet encoder learns space and structure, UNet decoder learns appearance and texture. With such a property, we design the forgetting mechanism by merging original checkpoint and optimized checkpoint to successfully tackle the fatal and inevitable overfitting issues when fine-tuning Diffusion Models on one image, thus significantly boosting the editing capability of Diffusion Models. Our method, Fast Imagic, even built on the outdated Stable Diffusion, achieves new state-of-the-art results on the challenging text-guided image editing benchmark: TEdBench, surpassing the previous SOTA methods such as Imagic with Imagen, in terms of both CLIP score and LPIPS score. Codes are available at `https://github.com/witcherofresearch/Forgedit`.

## 1 Introduction

Text-guided Image Editing (20) is a fundamental problem in computer vision, with a target text prompt indicating the editing intention to the given image. The approaches of text-guided image editing are generally categorized into optimization-based methods and non-optimization ones according to whether fine-tuning process is performed for reconstruction. Recent non-optimization editing methods (3; 31; 5; 33; 18; 1; 4; 32) are very efficient. Yet they either struggle on preserving the precise characteristics of original image during complex editing, or suffer from being incapable of performing sophisticated and accurate non-rigid edits. It is undeniable that fine-tuning a diffusion

---

[*]Or contact me via my personal email witcherofresearch@gmail.com. Codes are available at `https://github.com/witcherofresearch/Forgedit`.

model with the original image is still critical and necessary for high-precision identity preservation and accurate semantic understanding. However, previous optimization-based methods (14; 27) suffer from long fine-tuning time, severe overfitting issues or incapabliity of performing precise non-rigid editing. Here overfitting refers to the phenomenon that the model could reconstruct the original image, yet incapable of conducting the edit according to the target prompt, which we will demonstrate later.

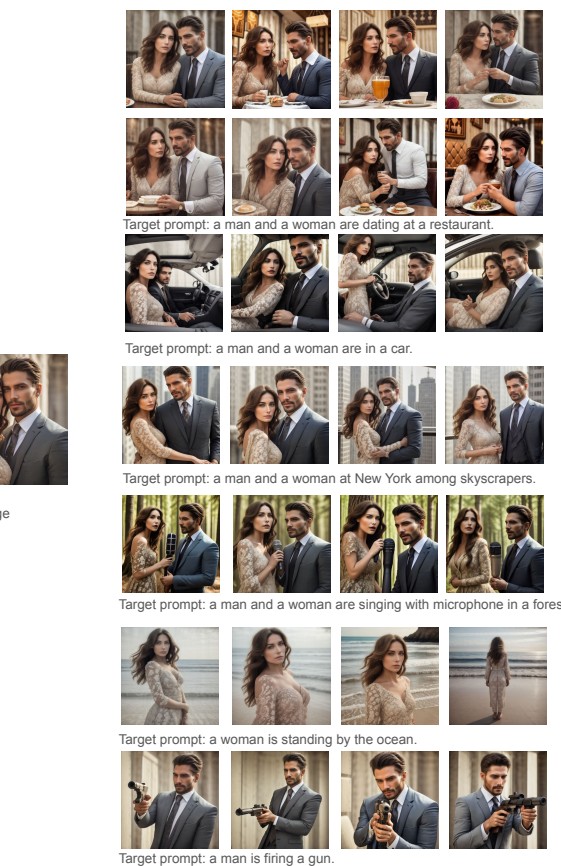

Input image

Target prompt: a man and a woman are dating at a restaurant.

Target prompt: a man and a woman are in a car.

Target prompt: a man and a woman at New York among skyscrapers.

Target prompt: a man and a woman are singing with microphone in a forest.

Target prompt: a woman is standing by the ocean.

Target prompt: a man is firing a gun.

Figure 1: Fast Imagic could be used for consistent and controllable keyframe generation for visual storytelling and movie generation, given one input image and target prompts. We list several samples with different random seeds for each target prompt. We demonstrate Fast Imagic is capable of controling multiple characters performing various actions at different scenes.Fast Imagic could also control each different character separately. Forgetting strategy on UNet's encoder with vector subtraction leads to high flexibility and success rate to change the spatial structures and actions, preserving appearance and identity by reserving UNet's decoder.

In this paper, we are going to tackle the aforementioned issues of the SOTA optimization-based editing method, Imagic (14). We name our text-guided image editing method Fast Imagic, which consists of two stages: fine-tuning and editing.

For fine-tuning stage, with a generated source prompt from BLIP (17) to describe the original image, we design a vision and language joint optimization framework, which could be regarded as a variant of Imagic(14) by combining the first stage and the second stage of Imagic into one and using BLIP generated caption as source prompt instead of using target prompt as source prompt like what Imagic does. Such simple modifications are the keys to much faster convergence speed and less overfitting than Imagic. With our joint learning of image and source text embedding, the finetuning stage using one image with our Fast Imagic+Stable Diffusion 1.4 (26) takes 30 seconds on an A100 GPU, compared with 7 minutes with Imagic +Stable Diffusion (14) reported by Imagic paper. This leads to 14x speed up. BLIP generated source prompt also eases overfitting, which we will demonstrate in the ablation study. In addition, our joint vision-language optimization eliminates the strange random flip phenomenon reported in Imagic, i.e., the direction of objects in the editing results randomly flip,

leading to editing failures. Our Fast Imagic always lock the correct direction of the objects the same as the original image, and only flips the directions of objects when the target prompt instructs so.

For editing stage, we propose two novel methods, vector projection in text embedding space and forgetting strategy with a finding of a general UNet disentangled property. For the first time in the literature of text-guided image editing with Diffusion Models, we propose a novel vector projection mechanism in text embedding space of Diffusion Models, which is capable to separately control the identity and editing strength by decomposing the language representations into identity embedding and editing embedding. We explore the properties of vector projection and compare it with previous vector subtraction method utilized in Imagic to demonstrate its superiority on identity preservation. Finally, we discovered a general property of UNet structure in Diffusion Models, i.e., UNet encoder learns space and structure, UNet decoder learns appearance and identity. With such a property, we could easily tackle the fatal overfitting issues of optimization-based image editing methods in a very effective and efficient manner during sampling process instead of fine-tuning process, by designing a forgetting mechanism with model merging according to UNet disentanglement. Without intention to reveal authors' information, Fast Imagic has been completely open-sourced for more than one year (of course, the open-sourced project is not called 'Fast Imagic', thankfully not violating the double-blind rule). We were the first to discover and open-source the encoder-decoder disentanglement phenomenon in Diffusion UNet models more than one year ago, though such a property is re-discovered in some recent papers.

To sum up, our main contributions are:
1. We present Fast Imagic, an efficient vision-language joint alignment framework, capable of performing both rigid and non-rigid text-guided image editing, while speeds up previous SOTA Imagic 14 times, completely solves the overfitting issue of Imagic.
2. We introduce a novel vector projection mechanism in text embedding space of Diffusion Models, which decomposes the target prompt representations into identity embedding and editing embedding. This improves Fast Imagic's capability for preserving more consistent characteristics of original image than existing methods.
3. We design a novel forgetting strategy via model merging based on our discovery on the disentangled UNet architecture of diffusion models, i.e., UNet encoder learns space and structure, UNet decoder learns appearance and texture. This allows us to effectively tackle the critical overfitting issue of optimization-based image editing methods, thus significantly boosting the editing capability of diffusion models.

Our Fast Imagic achieves new state-of-the-art results on the challenging benchmark TEdBench (14) (even by using an outdated Stable Diffusion 1.4), surpassing previous SOTA Imagic built on Imagen in terms of both CLIP score (8) and LPIPS score (34).

## 2  Related Works

**Test-time fine-tuning image editing** Diffusion Models have dominated text to image generation. DDPM(11) improves Diffusion process proposed by (29) on generating images. DDIM (30) accelerates the sampling procedure of Diffusion Models by making reverse process deterministic and using sub-sequence of time-steps. Dalle 2 (25) trains a diffusion prior to convert a text caption to CLIP (23) image embedding and then employs a Diffusion Decoder to transfer the generated CLIP image embedding to an image. Imagen (28) is a Cascaded Diffusion Model (12), whose UNet is composed of three Diffusion Models generating images with increasing resolutions, employing the powerful T5 text encoder (24) for complex semantic understanding and generating sophisticated scenarios. Stable Diffusion (26) utilizes Variational AutoEncoders (16) to compress the training image to a compact latent space so that the UNets could be trained with low resolution latents in order to save computational resources.These models are pretrained on billions of data. For image editing task with one given image, DreamBooth (27), textual inversion (6), Lora (13), Imagic(14) etc., could be trained with one image and conduct the edit with text to image generation.

**Test-time fine-tuning free image editing** There are some test-time finetuning-free methods, which do not require to optimize the diffusion model for each reference image. However, methods like SDEdit (18),DDIM inversion (30), MasaCtrl (4), Elite(32) all struggle to preserve the characteristics during complex editing and some of them could also change the view, pose and background irrelevant to target prompt, which leads to editing failures. Other typical methods like PnP Diffusion (31), Instruct Pix2pix (3), Prompt to Prompt (7) are incapable to conduct non-rigid editing and space-related editing.

Drag Diffusion (19), which extends DragGAN (21), is only capable of performing space-related editing, which is just a portion of general image editing tasks. Instead, our Fast Imagic is a general text-guided image editing framework to conduct various kinds of image editing operations, including spatial transformations.

## 3 Fast Imagic

### 3.1 Preliminaries

Diffusion models (11; 29) consist of a forward process and a reverse process. The forward process starts from the given image $x_0$, and then progressively add Gaussian Noise $\epsilon_t \sim \mathcal{N}(0, 1)$ in each timestep $t$ to get $x_t$. In such a diffusion process, $x_t$ can be directly calculated at each timestep $t \in \{0, ..., T\}$,

$$x_t = \sqrt{\alpha_t}x_0 + \sqrt{1 - \alpha_t}\epsilon_t \tag{1}$$

with $\alpha_t$ being diffusion schedule parameters with $0 = \alpha_T < \alpha_{T-1}... < \alpha_1 < \alpha_0 = 1$ .

In the reverse process, given $x_t$ and text embedding $e$, the time-conditional UNets $\epsilon_\theta(x_t, t, e)$ of diffusion models predict random noise $\epsilon_t$ added to $x_{t-1}$. With DDIM (30), the reverse process can be computed as,

$$x_{t-1} = \frac{\sqrt{\alpha_{t-1}}}{\sqrt{\alpha_t}}(x_t - \sqrt{1 - \alpha_t}\epsilon_\theta(x_t, t, e)) + \sqrt{1 - \alpha_{t-1}}\epsilon_\theta(x_t, t, e) \tag{2}$$

With Latent Diffusion Models (26), the original image $x_0$ is replaced by a latent representation $z_0$ obtained from a VAE (16) Encoder $\varepsilon(x_0)$. The overall training loss is computed as,

$$L = \mathbb{E}_{z_t, \epsilon_t, t, e}||\epsilon_t - \epsilon_\theta(z_t, t, e)||_2^2 \tag{3}$$

### 3.2 Joint vision-language optimization for alignement

In order to tackle such challenging text-guided image editing problems, we propose a image and text alignment framework via joint optimization of text embedding and UNet with the given image. Shown in Figure 2, we introduce the overall design of our vision-language joint optimization framework.

**Source prompt generation.** We first use BLIP (17) to generate a caption describing the original image, which is referred to as the source prompt. The source prompt is then fed to the text encoder of Stable Diffusion (26), generating an embedding $e_{src}$ of source prompt. Previous three-stage editing method Imagic (14) regards target prompt text embedding as source one $e_{src}$. We found that it is essential to use the BLIP caption instead of using the target prompt as a pseudo source prompt like Imagic. Otherwise such fine-tuning methods easily lead to overfitting issues, as demonstrated in the 5th column 'Imagic SD' of Figure 6. This phenomenon indicates that using the BLIP caption as source prompt would result in better semantic alignment with the given original image than Imagic.

**Vision-language alignment with joint optimization.** We choose to optimize UNet encoder blocks of 0, 1, 2 and decoder blocks of 1, 2, 3 in the UNet structure since we found that fine-tuning deepest features would lead to overfitting in our Fast Imagic framework, demonstrated in Figure 2. Similar with Imagic, we regard source text embedding as parameters to optimize. Yet different with Imagic which optimizes text embedding and UNet in two separate stages, we found it vital to align the source text embedding and UNet parameters simultaneously, which is of great importance for faster convergence and better reconstruction quality than Imagic. In particular, due to a large domain gap between text and image, we use different learning rates for source text embedding ($10^{-3}$) and UNet ($6 \times 10^{-5}$) with Adam Optimizer (15). For faster training, since we only have a single training image, we repeat the tensors on batch dimension for batch-wise optimization with a batch size of 10. We use mean square error loss, and empirically found that stable reconstruction results can be achieved when the final loss is less than 0.03. With the batch size set to 10, the models are fine-tuned for 35 to 40 steps. We stop the training over 35 steps when the loss is less than 0.03, or stop at 40 steps at most. This fine-tuning process is significantly more efficient than Imagic, taking 30 seconds on a single A100 GPU. The training loss is computed as,

$$L = \mathbb{E}_{z_t, \epsilon_t, t, e_{src}}||\epsilon_t - \epsilon_{\theta, e_{src}}(z_t, t, e_{src})||_2^2 \tag{4}$$

where the main difference with the training loss presented in 3 is that $e_{src}$ is considered as parameters to optimize.

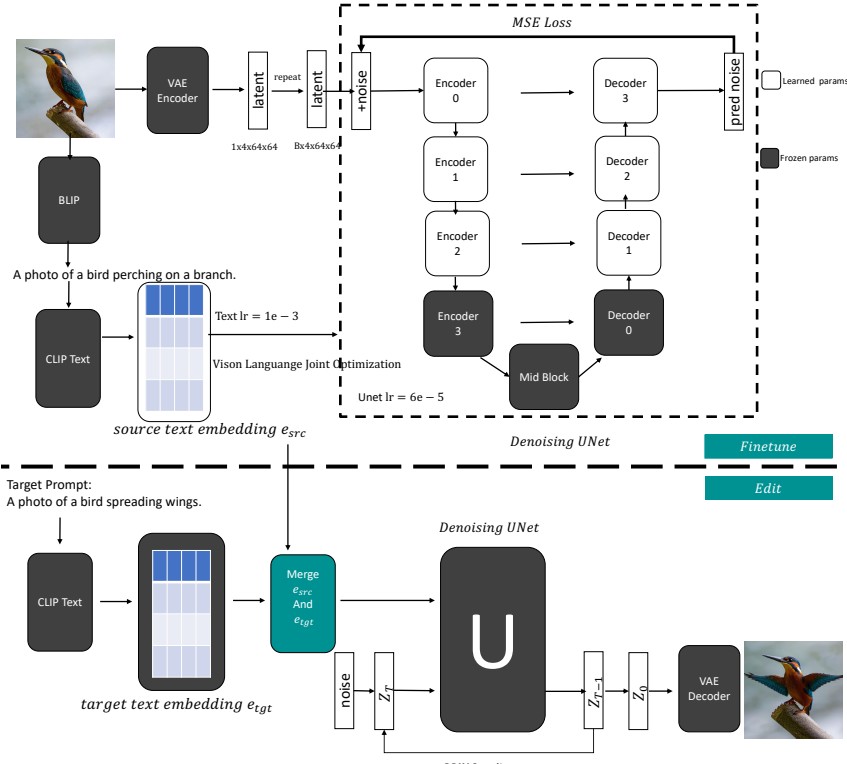

Figure 2: Overall framework of our Fast Imagic, consisting of a vision-language joint fine-tuning stage and an editing stage. We use BLIP to generate a text description of an original image, and compute an embedding of the source text $e_{src}$ using a CLIP text encoder. The source embedding $e_{src}$ is then jointly optimized with UNet using different learning rates for text embedding and UNet, where the deep blocks of UNet are frozen. During the editing process, we merge the source embedding $e_{src}$ and the target embedding $e_{tgt}$ with vector subtraction or projection to get a final text embedding $e$. With our forgetting strategies applied to UNet, we utilize DDIM sampling to get the final edited image.

## 3.3 Reasoning and Editing with language representation decomposition

We first input the target prompt to the CLIP (23) text encoder of the Stable Diffusion model (26), computing a target text embedding $e_{tgt}$. With our learned source text embedding $e_{src}$, we introduce two methods to combine $e_{src}$ and $e_{tgt}$ so that the merged text embedding can instruct the UNet to preserve characteristics of original image and also follow the target prompt. Given $e_{src} \in \mathbb{R}^{B \times N \times C}$ and $e_{tgt} \in \mathbb{R}^{B \times N \times C}$ , we conduct all vector operations on the $C$ dimension to get the final text embedding $e$.

**Vector Subtraction.** We use the same interpolation method as Imagic (14),

$$e = \gamma e_{tgt} + (1 - \gamma)e_{src} = e_{src} + \gamma(e_{tgt} - e_{src}) \tag{5}$$

As shown in Figure 3, the final text embedding $e$ is obtained by travelling along vector subtraction $e_{tgt} - e_{src}$ . In our experiments, we found that in most cases, $\gamma$ goes beyond 1 when the editing is performed successfully. This leads to a problem that the distance between the final embedding $e$ and the source embedding $e_{src}$ may be so far that the appearance of the edited object could change vastly.

**Vector Projection.** We propose to use vector projection to better preserve the appearance of the original image. As shown in the Figure 3, we decompose a target prompt text embedding $e_{tgt}$ into a vector along $e_{src}$ and a vector orthogonal to $e_{src}$. We call the orthogonal vector $e_{edit}$. We first calculate the ratio $r$ of the projected vector on $e_{src}$ direction.

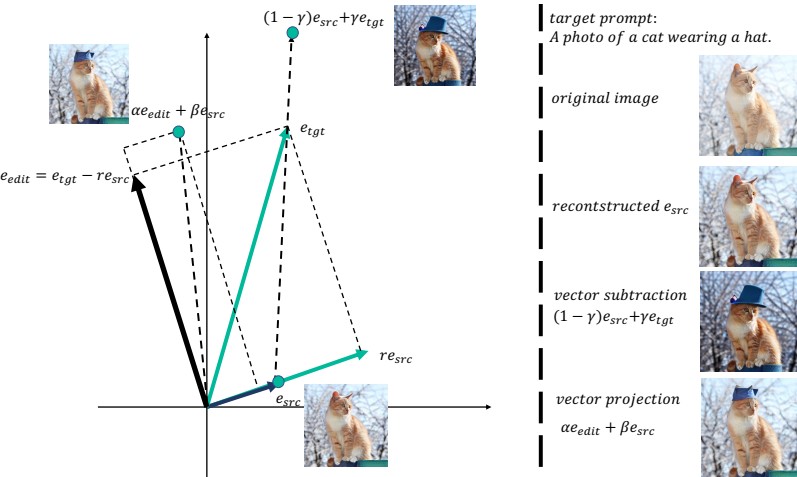

Figure 3: We demonstrate vector subtraction and vector projection to merge $e_{src}$ and $e_{tgt}$. Vector subtraction could lead to inconsistent appearance of the object being edited since it cannot directly control the importance of $e_{src}$. The vector projection decomposes the $e_{tgt}$ into $re_{src}$ along $e_{src}$ and $e_{edit}$ orthogonal to $e_{src}$. We can directly control the scales of $e_{src}$ and $e_{edit}$ by summation.

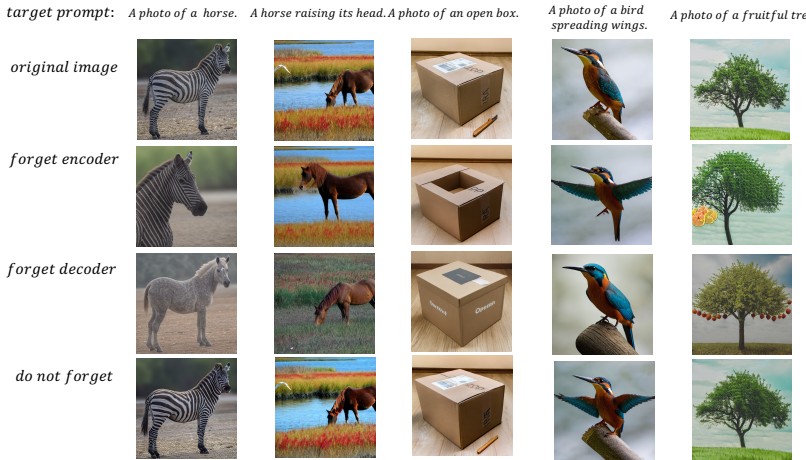

Figure 4: The encoder of UNets learn features related to pose, angle, structure and position. The decoder are related to appearance and texture. Thus we design a forgetting strategy according to the editing target.

$$r = \frac{e_{src}e_{tgt}}{||e_{src}||^2} \tag{6}$$

Thus, we could get the $e_{edit}$ by computing

$$e_{edit} = e_{tgt} - re_{src} \tag{7}$$

To control the characteristics similarity and editing strength separately, we sum $e_{src}$ and $e_{edit}$ with two coefficient $\alpha$ and $\beta$,

$$e = \alpha e_{src} + \beta e_{edit} \tag{8}$$

**Editing.** We use DDIM sampling (30) with a classifier free guidance (10) to conduct the edit. The guidance scale is 7.5. For vector subtraction, we iterate over a range of $\gamma \in [0.8, 1.6]$. For vector projection, we choose $\alpha$ from two values $\{0.8, 1.1\}$, and $\beta$ from a range of $[1.0, 1.5]$.

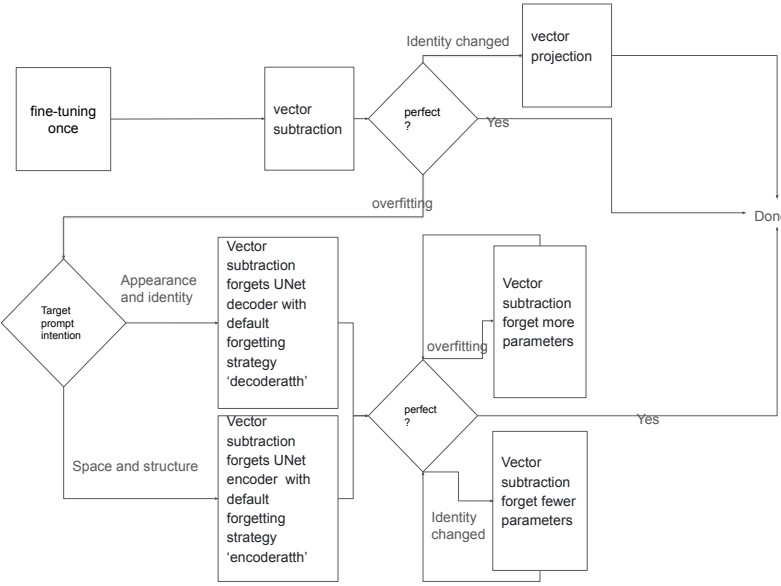

Figure 5: Fast Imagic Workflow.

### 3.4 The Ultimate Solution to overfitting with disentangled UNet and forgetting mechanism

**Forgetting mechanism** Considering the fact that there is only one training image provided, in some cases the diffusion model still overfits, thus losing the editing capability, though joint vision-language alignment could ease the overfitting to some extent. The fine-tuning process is computational expensive compared to sampling process, thus we design a novel forgetting mechanism during sampling process to tackle the overfitting problem. The network is only fine-tuned once, and can be converted to multiple different networks during sampling process by merging certain fine-tuned parameters $w_{learned}$ and the corresponding parameters of original UNet (before fine-tuning) $w_{orig}$, with a balance coefficient $\sigma$. In practice, we found that $\sigma = 0$ works in general, which means that we can simply replace the learned parameters with original parameters so that the network completely forgets these learned parameters. However, which paramters should be forgotten?

$$w = \sigma w_{learned} + (1 - \sigma)w_{orig} \tag{9}$$

**Disentangled UNet** Shown in Figure 4, we found an general disentanglement property of UNet in diffusion models. The encoder of UNets learns space and structure information like the pose, action, position, angle and overall layout of the image, while the decoder learns appearance and textures instead. We were the first to draw a clear and universal conclusion on disentangled UNet and open-sourced our solution completely more than one year ago. In Figure 4 , given the target prompt and original image in the first row, we conduct Fast Imagic with forgetting UNet encoder, UNet decoder and nothing in the second, third and fourth rows respectively. We could see that without forgetting mechanism, four out of five cases are overfitting. By forgetting UNet encoder, the structure and space features are changed yet the appearance and texture are preserved. Vice versa for forgetting UNet decoder.

**Tackling overfitting with disentangled UNet and forgetting mechanism** If the target prompt tends to edit space and structure information, for example, the pose or layout, we will choose to forget parameters of the encoder. If the target prompt aims to edit the appearance, the parameters of decoder should be forgotten. Currently we only apply the forgetting strategy when a text embedding $e$ is obtained by vector subtraction in previous section. We will conduct a thorough exploration of forgetting mechanism with disentangled UNet in appendix due to page limit.

### 3.5 WorkFlow

The overall workflow of Fast Imagic is explained in Figure 5. The fine-tuning stage is the same for all images. The diamonds in the figure indicate that the process depends on the users's choices and preferences. In practice, these user decisions can also be replaced by thresholds on metrics like CLIP

| Editing method | CLIP Score ↑ | LPIPS Score ↓ | FID Score ↓ |
|---|---|---|---|
| Imagic+Imagen (14) | 0.748 | 0.537 | 8.353 |
| Fast Imagic+SD (ours) | **0.771** | **0.534** | **7.071** |

Table 1: Our Fast Imagic with Stable Diffusion is the new state-of-the-art text-guided image editing method on the challenging benchmark TEdBench, surpassing previous SOTA Imagic+Imagen.

score and LPIPS score, for completely automatic editing. Due to page limit, we will demonstrate the workflow with practical examples in appendix 13.

## 4    Experiments

**Benchmark.** TEdBench (14) is one of the most difficult public-available text-guided image editing benchmarks. It contains 100 editings, with one target prompt and one image for each edit. These target prompts are very general with diversity, including but not limited to changing the appearance of objects, replacing certain parts of the image, changing the position, action and number of the object, editing multiple objects with complex interactions. In particular, non-rigid edits turn out to be very tough for many SOTA text-guided image editing methods. In terms of quantitative evaluation, we utilize CLIP Score (8) to measure semantic alignments with target prompt, and LPIPS score (34) and FID score (9) to indicate fidelity to the original image.

### 4.1    Ablation Study

**joint vision-language alignment.**   We explore the importance of using generated caption as source prompt and joint alignment. We use BLIP generated source prompt to describe the original image, yet previous SOTA method Imagic uses target prompt as source prompt. Since target prompt indicates the editing target, it is obviously inconsistent with the original image. In Figure 11, we show cases that do not need to use forgetting strategy from Figure 6, so that we could remove the effects of forgetting strategy. If target prompt is used instead of BLIP generated source prompt, all these cases of Fast Imagic without using generated source prompt will overfit.

We could also find it vital to align vision and language simultaneously instead of in separate stages in order to ease overfitting, by comparing 'Imagic SD' and 'Fast Imagic SD' columns in Figure 11 since all the cases in this figure do not use forgetting mechanism.

**decomposition in text embedding space.** We compare two different reasoning methods to merge $e_{src}$ and $e_{tgt}$ to get the final text embedding $e$, shown in Figure 7 . These two methods are complementary to each other, with vector projection better at preserving the identity, and vector subtraction showing stronger editing capability. Thus in the workflow of Figure 5, we use vector subtraction to be the default option. When the characteristics could not be preserved by vector subtraction, we switch to vector projection.

**Disentangled UNet.** We explore what effects could the location and amount of forgotten parameters cause to the editing results, in terms of UNet encoder in Figure 9 and UNet decoder in Figure 8. The default strategies are 'encoderattn' and 'decoderattn' in Figure 5. In fact, our findings on disentangled UNet could explain several phenomenons in controllable generation and image editing.
1. Why ControlNets (33) copy the UNet encoder branch instead of UNet decoder? Because UNet encoder controls the space and structure of generated image. ControlNet utilize extra condition to fix the layout and structure and does not control the texture.
2. Why methods like PnP diffuion (31) could not conduct non-rigid editing, for example, change the pose and action of objects? because they only operate on UNet decoder and fix the UNet encoder, which means that the space and structure cannot be edited.

**hyperparameters effects for text embedding interpolation** We explore the effects of hyperparameters in vector subtraction and vector projection in Figure 10 and in Figure 13.

### 4.2    Comparison with State-of-the-art

We compare qualitative editing results of our Fast Imagic with SOTA methods on several random test samples from TEdBench in Figure 6, demonstrating stronger semantic alignments with target prompts and more precise identity preservation than other methods. Quantitatively, our Fast Imagic with the even outdated Stable Diffusion 1.4, surpasses the current SOTA Imagic+Imagen on TEdBench benchmark in terms of both CLIP Score, LPIPS Score and FID Score, shown in Table 1. For FID

score, we follow the advice of the authors by setting dimension to 192 since there are only 100 samples in TEdBench.

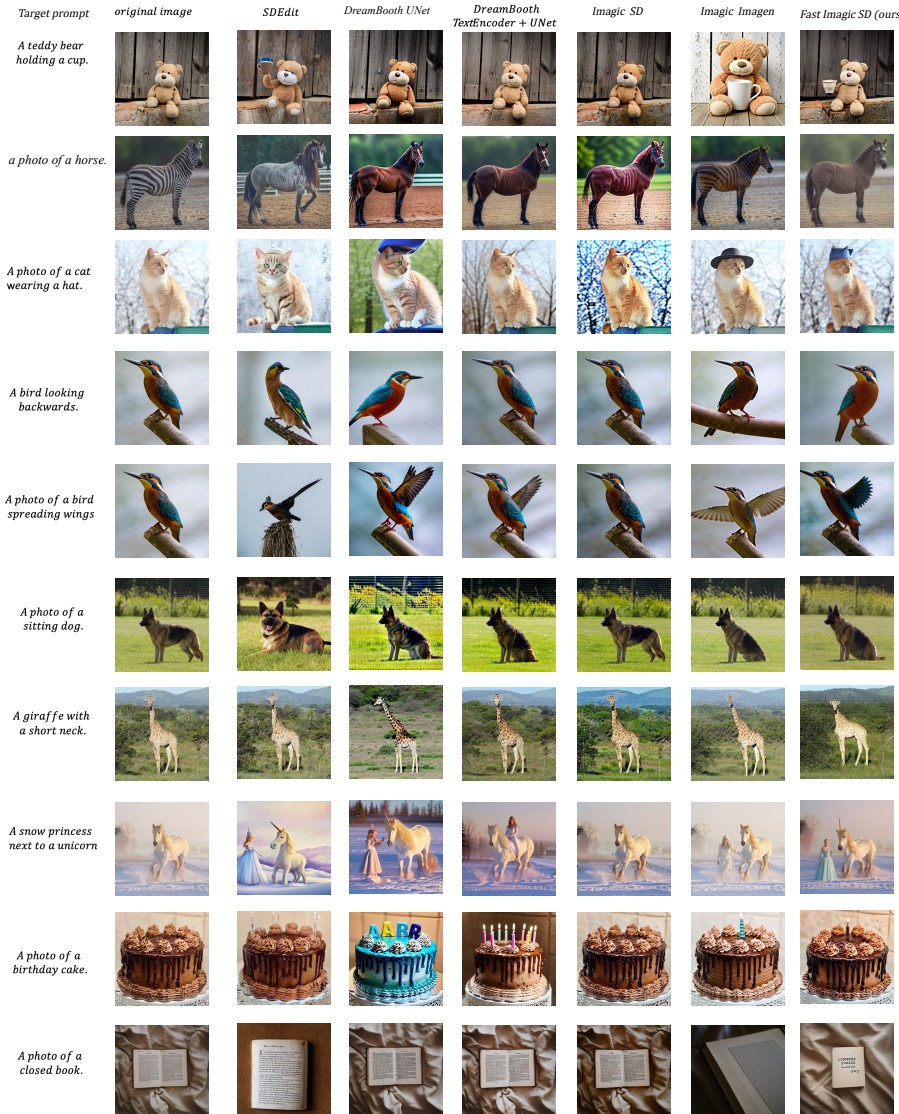

Figure 6: Comparison with SOTA: non-optimization SDEdit, optimization BLIP+DreamBooth and Imagic, demonstrating the strong editing ability and stable identity preservation of Fast Imagic.

## 5 Conclusion

We present our Fast Imagic framework to tackle the challenging text-guided image editing problem. Fast Imagic speeds up previous SOTA Imagic by 14 times, and completely solves the overfitting problem of Diffusion Models when fine-tuning with only one image, via vision-language joint alignment and disentangled UNet with forgetting mechanism, and obtain new SOTA on TEdBench.

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

# 6 Appendix / supplemental material

## 6.1 Visual Storytelling

Our Fast Imagic could precisely preserve the characteristics of multiple actors and is capable of conducting complex non-rigid editing, which makes our Fast Imagic an ideal tool for visual storytelling and long video generation with strong consistency and very arbitrary scene and action. In Figure 1, we input a random image generated by SDXL (22) and then use Fast Imagic with Realistic Vision V6.0 B1 noVAE, a variant of Stable Diffusion to generate various samples for different target prompts. With image to video models, for example Stable Video Diffusion (2), we could generate movies with high consistency of several minutes.

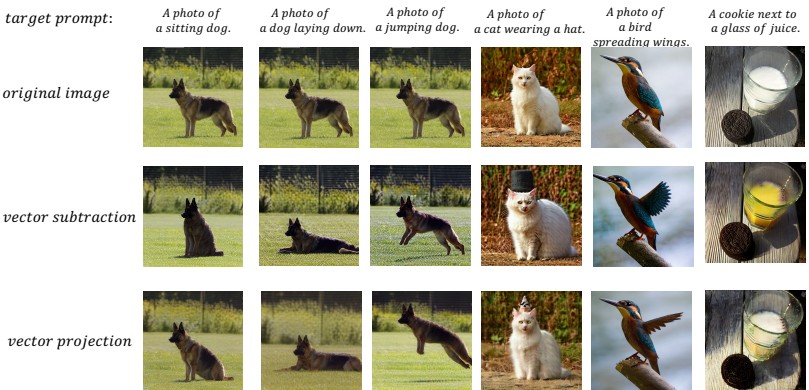

Figure 7: Comparisons of vector subtraction and vector projection, which are complementary.

## 6.2 decomposition in text embedding space

We compare two different reasoning methods to merge $e_{src}$ and $e_{tgt}$ to get the final text embedding $e$, shown in Figure 7 . These two methods are complementary to each other, with vector projection better at preserving the identity, and vector subtraction showing stronger editing capability. Thus in the workflow of Figure 5, we use vector subtraction to be the default option. When the characteristics could not be preserved by vector subtraction, we switch to vector projection.

For the dog and the cat examples, vector projection can preserve more details of the appearance of the dog and the cat than vector subtraction. However, for a glass of milk and cookie example, vector subtraction performs better than vector projection which struggles to change the milk to juice and also introduces wave-like blurs in the image. We observe such phenomenons in many other cases for vector projection, which demonstrates that it is more suitable for edits where the identity of object should be kept instead of changed.

## 6.3 Disentangled UNet

The default workflow is in the highlighted flow in Figure 12, where the default forgetting mechanisms are 'encoderattn' and 'decoderattn'. However, we still demonstrate in Figure 9 and Figure 8 how changing which part of parameters to merge the models influence the editing result. We first inference without forgetting strategies. If overfitting happens, we choose from the default 'encoderattn' or 'decoderattn' strategy according to the UNet property and target prompt intention. The 'encoderattn' means forgetting all encoder parameters except attention-related parameters. 'decoderattn' means forgetting all decoder parameters except attention-related parameters. The user may choose to forget more or fewer parameters according to the editing results, which we demonstrate and explain in 9 and 8.

## 6.4 The interpolation hyper-paramters of vector subtraction and vector projection

We explore the effect of hyper-parameters to the editing result in Figure 10.

## 6.5 Joint optimization of vision and language

The fine-tuning process of Imagic is composed of two stages, text embedding optimization for 500 steps and UNet optimization for 1000 steps. Our Fast Imagic employ unified vision language optimization with a batch-wise traing on 1 a100 GPU, which leads to 40 steps in total, speeding up Imagic for 14 times. This unified optimization of vision and lanugage goes beyond speeding up Imagic. It also eases the overfitting issue to some extent, shown in Figure 11.

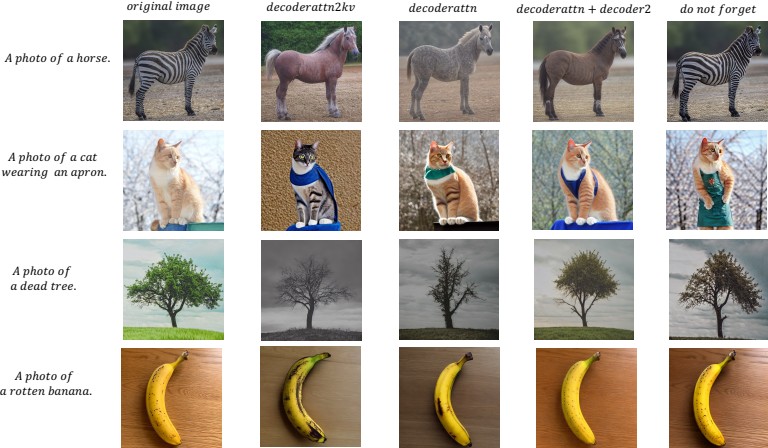

Figure 8: We explore various forgetting strategies for decoder. All learned encoder parameters are preserved. In the $2^{nd}$ to $4^{th}$ columns, we preserve decoder cross-attention parameters, decoder self-attention and cross-attention, decoder self-attention, cross-attention and the entire decoder2 block, forgetting all the other parameters of decoder.

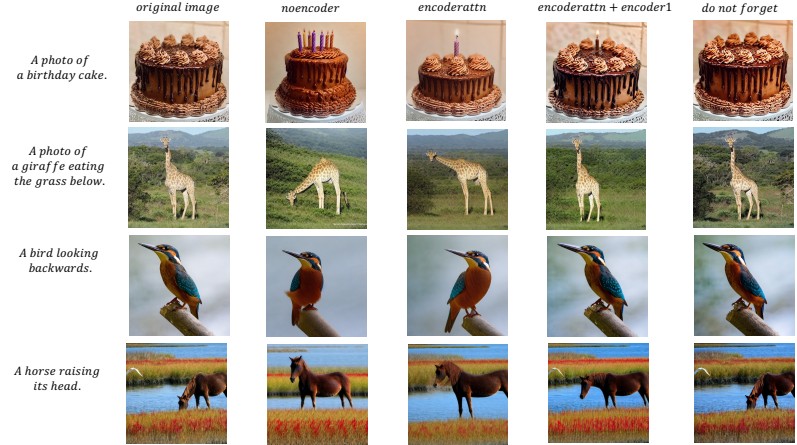

Figure 9: We explore different forgetting strategies for encoder. All learned decoder parameters are preserved. For the second to fourth column each, we preserve none of the encoder parameters, encoder self attention and cross attention, encoder self attention and cross attention and the entire encoder1 block, forgetting all the other parameters of encoder.

## 6.6 Practical Examples of the Workflow

The overall workflow is shown in Figure 12. No matter what the input image is, we use the same set of hyper-paramters for finetuning stage. In the editing stage, the default workflow is to use vector subtraction with $\gamma$ in the range of 0.8 to 1.6. In general, a proper editing result should already been obtained from one of these 8 images. However, if a perfect editing did not show up, there are two possibilities, overfitting or underfitting. Underfitting leads to the fact that the edited object suffers from identity shift, which means with the editing strengthened, the appearance of target object becomes gradually inconsistent with input image. In this circumstance, one needs to apply vector projection instead, which I will show in another paper with examples from TEdBench. The more often case is overfitting, which means that Fast Imagic could reconstruct the input image well yet cannot conduct the edit successfully. With the disentangled property of UNet, we could utilize the forgetting strategy to tackle the overfitting issue. If the target prompt aims to edit space and structure, one should use the default "encoderattn" forgetting strategy. If the target prompt aims to edit appearance

*original image*

*target prompt*:  *A photo of a cat wearing a hat.*

*vector subtradtion*    $(1-\gamma)e_{src}+\gamma e_{tgt}$

$\gamma = 0.6\ to\ 1.5$

*vector projection*    $\alpha e_{edit} + \beta e_{src}$

$\beta=0.7$

$\alpha = 1.0\ to\ 1.6$

$\beta=0.9$

Figure 10:  $\gamma$ for vector subtraction and $\alpha$, $\beta$ for vector projection.

and texture, one should use the default "decoderattn" forgetting strategy. Using the examples from EditEvalv1 benchmark, we demonstrate several cases on how to adjust the hyper-parameters. The base model used in the following examples is Stable Diffusion 1.4.

For the first case where the input image is a polar bear on the ice field, the target prompt is "A polar bear raising its hand". To begin the workflow in Figure, we first run vector projection without forgetting strategy with with $\gamma$ in the range of 0.8 to 1.6. Shown in Figure , we could find that we are facing the overfitting issue and the polar bear is incapable of raising its hands. Following Figure , we then run the default forgetting strategy on UNet's encoder, i.e. "encoderattn", which means that newly learned parameters of self attention blocks and cross attention blocks are preserved in UNet encoder and all learned parameters of UNet decoder are preserved as well. The hyper-parameter $\gamma$ still ranges from 0.8 to 1.6. This time we could find successful edits in the results.

## 6.7   Limitations

First the effect of Fast Imagic is influenced by randomness. The fine-tuning process inevitably introduces randomness thus for some particular cases, we cannot guarantee to perfectly reconstruct the details of original image thus we have to run the fine-tuning stage several times for these challenging cases. The sampling procedure is also related to the initial random seed of reverse process, thus for some extremely challenging cases we have to sample tens of images or even hundreds, though rarely the case, before we could get a proper edited one.

Second, the editing capability of Fast Imagic is restricted by the utilized Diffusion Model. If the target prompt cannot even be generated by the Diffusion Model itself, it is almost impossible to accomplish the edit according to the target prompt. For example, the prompt 'a sitting flamingo'

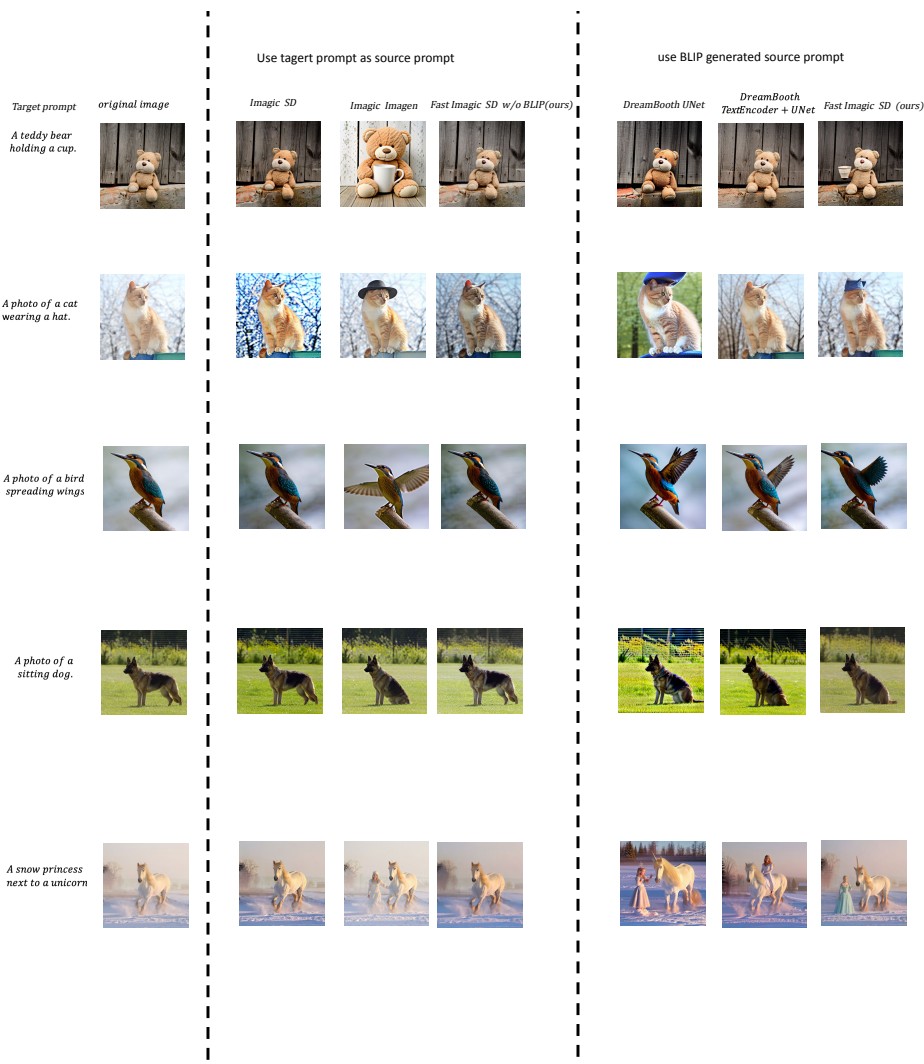

Figure 11: What should the source prompt be? Excluding the usage of forgetting strategies for ablation, we could find that Fast Imagic using target prompt leads to severe overfitting, yet Fast Imagic using BLIP generated source prompt eases overfitting.

cannot be generated by Stable Diffusion at all, thus Fast Imagic cannot successfully edit it either. Such an issue could possibly be solved by switching to better Diffusion Models.

We show some typical bad cases in Figure 14.

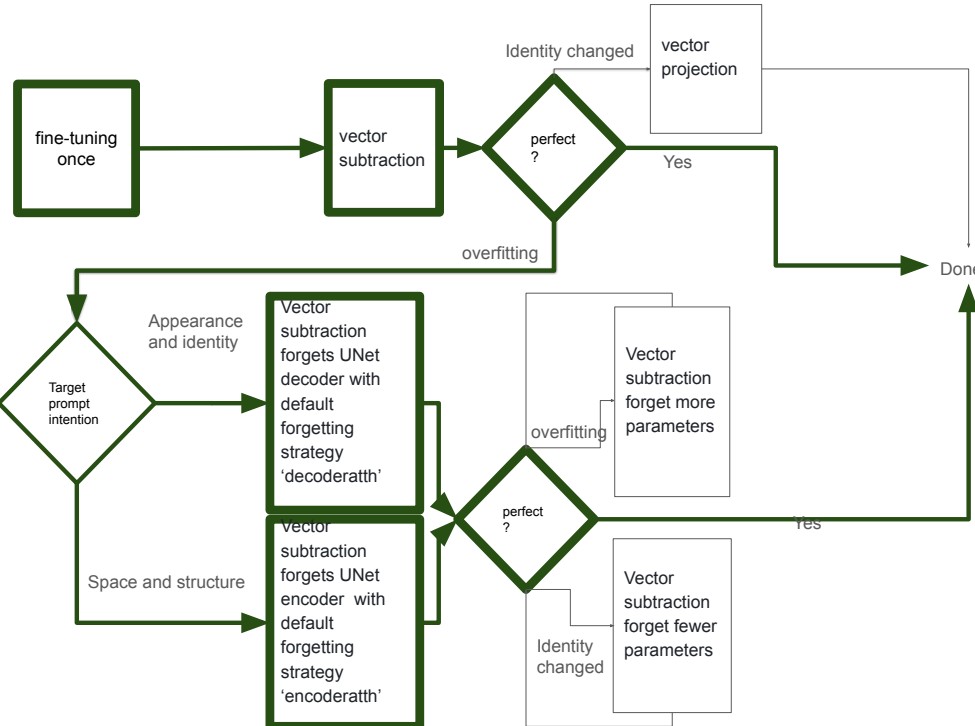

Figure 12: The workflow of Fast Imagic, the most usual flow of editing process is highlighted in the figure, i.e. simple vector subtraction and default forgetting strategies according to our findings of the disentangle rules of UNet.

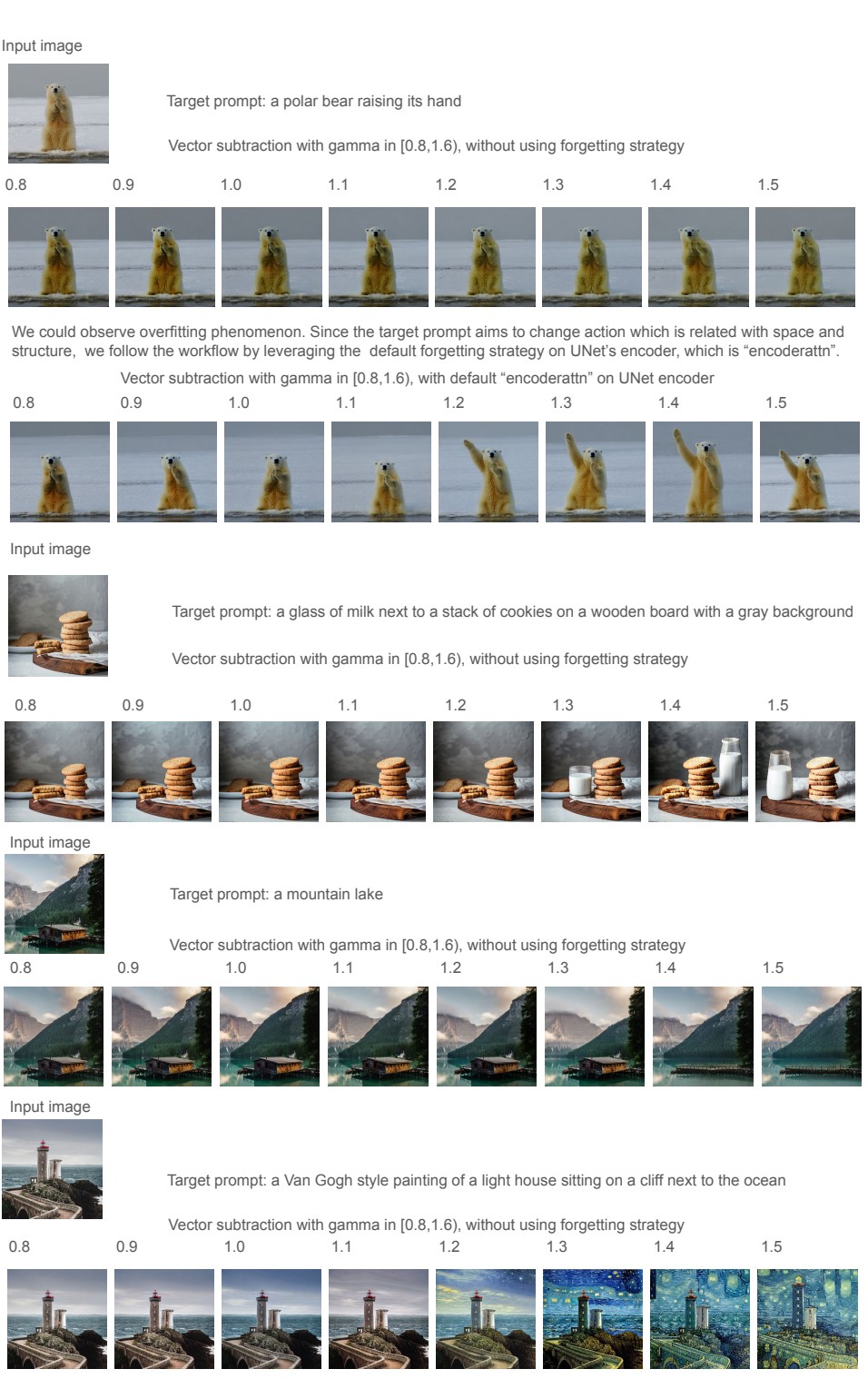

Figure 13: We show the practical workflow of Fast Imagic, with testing images from EditEval. In most cases, simple vector subtraction would finish the job. For other hard cases, the default forgetting strategies, 'encoderattn' or 'decoderattn' according to editing intention on structrue or appearance, could solve the problems.

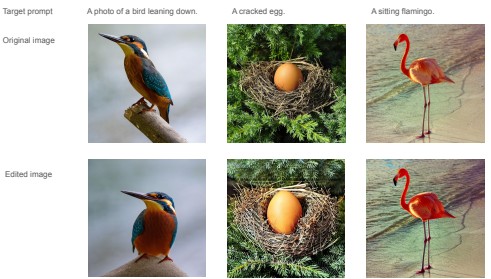

Figure 14: Bad cases from TEdBench.

