# OpenReview forum: "Fast Imagic: Solving Overfitting in Text-guided Image Editing via Disentangled UNet with Forgetting Mechanism and Unified Vision-Language Optimization"
_NeurIPS.cc/2024/Workshop/UniReps — UniReps_

### Official Review · Reviewer_RrvN · 2024-09-30
**Fast Imagic offers a quick and effective approach to text-guided image editing, solving issues like overfitting and slow performance, but it could use more examples, extra benchmarks, and a deeper dive into how each new feature contributes.**

**Rating:** 7
**Confidence:** 2

**Review:**

Strenghts:
1. The 14x speed-up in optimisation compared to Imagic is an impressive result. Clever idea to merge the first two stages of Imagic into one and using BLIP-generated captions.
2. The authors convincingly argue that by separating the UNet encoder and decoder responsibilities (space/structure vs. appearance/texture), they can overcome the typical overfitting issues that occur with fine-tuning on a single image.
3. The vector projection technique introduced for separately controlling identity and editing strength in the text embedding space is an insightful addition.

Weaknesses:
1. Limited qualitative examples. The paper presents only a few qualitative examples. Give the lack of examples in used dataset, extending this to show its application in diverse domains, such as fashion visual try-ons, could be possible solution to further highlight the method’s versatility.
2. Evaluation is performed on just one benchmark (TEdBench) and quantitative evaluation is done only against one method. Including more datasets or comparisons against a wider range of methods would provide a stronger validation of the approach’s generalization and robustness.
3. Limited quantitative ablation studies - while the paper highlights new techniques like joint optimization, vector projection, and the forgetting mechanism, it lacks a thorough quantitative ablation study to evaluate the individual contributions of these methods. This would provide clearer insight into how each component affects performance.
4. Although authors propose threshold approach, method requires some degree of manual effort.
5. Minor typographical errors: “capablity” (line 178) and “seperately” (line 170).

---

### Official Review · Reviewer_RdmH · 2024-10-02
**Very good paper with minor flaws.**

**Rating:** 8
**Confidence:** 4

**Review:**

### Summary
In this paper, the authors presented a novel framework for image editing using Diffusion Models. In particular, the authors put emphasis on solving the problems with overfitting and high computation time. The presented method proved to be significantly superior to the baseline solution, both in terms of efficiency and inference time.

### Major strengths:
* The authors performed an in-depth analysis of the problem and their method.
* The paper is well-structured and easy to read.
* The authors provided very interesting observations such as disentanglement in UNet or using BLIP captions to reduce overfitting.
* The paper was designed with the usability of the presented method in mind, giving the reader many hints for how to utilize it effectively.
* The experiments proved the effectiveness of the proposed method.

### Major weaknesses:
* The authors did not perform quantitative analysis using reference methods other than Imagic + Imagen, despite performing the qualitative analysis using them.

### Minor weaknesses:
* In Figure 5, the arrows showing the flow are barely visible, the font varies in size and the whole thing is quite chaotic. In Figure 12, the bolding obscures some of the text. The whole thing looks quite amateurish and does not match the rest of the paper.

---

### Official Review · Reviewer_iNgt · 2024-10-04
**Review Fast Imagic.**

**Rating:** 6
**Confidence:** 3

**Review:**

**Summary**
The authors propose a variation of the existing Imagic framework for text-guided image editing. The main contributions are:
1. Simultaneously optimize the text embedding and the UNet rather than in two stages.
2. Utilize BLIP captions as starting point for text embedding optimization rather than the target text
3. Explore alternative text embedding projection at inference time.
4. Design a UNet forgetting strategy to avoid overfitting in the Imagic framework

**Strengths and Weaknesses**

*Strengths*
1. The paper shows SOTA performance in the TEdBench
2. The paper is easy to follow and understand, it is well written and organized, with a few unclear details.
3. The paper explores novel ideas on the properties of denoising UNets in diffusion models, using BLIP for captioning and jointly optimizing text embedding and unet in Imagic.

*Weaknesses*
1. Despite showing SOTA performance, the TEdBench is a small benchmark of 100 image-text pairs.
2. Unlike Imagic, there is no user study backing the CLIP Score, LPIPS Score and FID Score. It is hard to tell if they are statistically significant as presented in the paper, given the size of the dataset.
3. The interesting ideas presented are not backed by **quantitative** ablation studies and rely on anecdotal examples. All together it is hard to assess which ones of the contributions is actually impacting the performance.
4. It is unclear what configuration of the method is used to produce the SOTA metrics reported.
5. Quantitative results only include Imagic and no other methods like SDEdit and others.
6. Even though it appears that the presented method is much faster than the original Imagic (30s vs 7m) it is unclear if the gains are driven by the adoption of BLIP embeddings or the joint optimization of embeddings and unet or both. Having an ablation study on this would be better.

**Questions**
1. There is a claim in the original Imagic paper that getting a resulting embedding after optimization close to the target embedding is critical for a seamless interpolation between original an target embeddings:
*We run this process for relatively few steps, in order to remain close to the initial target text embedding, obtaining eopt. This proximity enables meaningful linear interpolation in the embedding space, which does not exhibit linear behavior for distant embeddings.*
Do you not observe this need at all since you propose utilizing BLIP instead?
2. What configuration was used to produce the SOTA results reported? In particular, what forgetting mechanism? What embedding interpolation method? Were they the same across samples or some version of the workflow in Figure 5 was used? And, if Fig 5 was used, was it automated? how?
3. The original Imagic paper presents vastly different training configuration (100 steps for embedding optimization and 1500 for unet optimization). Do you think this performance was optimized? Were you able to replicate the experiment with a smaller number of steps?
4. Given that the editing task is quite fine-grained and often involves small detail changes, CLIP Score and LPIPS seem weaker metrics than a user study. Can you provide a justification for the use of these metrics, which are not the central metrics reported previous SOTA work such as Imagic?

**Soundness**: 2

**Presentation**: 3

**Contribution**: 2

**Overall**: 6

---

### Official Review · Reviewer_Xbt2 · 2024-10-06
**Comprehensively explained engineering solution to improve on the SoTA with interesting insights.**

**Rating:** 7
**Confidence:** 4

**Review:**

The work is focused on the setting of natural language based image editing programs which are fine-tuned on a given image to achieve granular control over how the image is edited. The paper provides insights into how the authors beat the SoTA for this particular setting i.e. Imagic, that deepen our understanding of how various components is such VLM systems behave.

There are multiple grammatical errors throughout the paper that detract from the high quality of the work itself. The appendix, which the reviewer is not required to read, is essential to the paper. Particularly the decomposition and disentanglement should have been further explored in the main paper as they are the most valuable insights in the reviewer's opinion.. Overall the paper feels like a compilation of insights organically gleamed from trying to engineer improvements on Imagic and less like an effort fundamentally understand or change the entire system.

Besides the above criticisms, the paper comprehensively explained everything. It clearly explained the setting of the problem, compared itself to multiple other works and justified its novelty and performance enhancements more than satisfactorily. It experimentally demonstrated  that it beat the SoTA and sufficiently proved its claimed contributions, which are impressive.

---

### Decision · Program_Chairs · 2024-10-10

**Decision:**

Accept

**Comment:**

In light of the positive reviewers' feedback and relevancy of the submission, we are pleased to accept this paper for presentation at UniReps 2024. We kindly ask the authors to incorporate the reviewers' suggestions and feedback in the final camera-ready version of the manuscript.